# Fibrosis growth factor 23 is a promoting factor for cardiac fibrosis in the presence of transforming growth factor-β1

Kazuhiro Kuga[1], Yoichiro Kusakari[1], Ken Uesugi[1,2], Kentaro Semba[2], Takashi Urashima[3], Toru Akaike[1], Susumu Minamisawa[1,2]*

**1** Department of Cell Physiology, The Jikei University School of Medicine, Tokyo, Japan, **2** Department of Life Science and Medical Bioscience, Waseda University, Tokyo, Japan, **3** Department of Pediatrics, The Jikei University School of Medicine, Tokyo, Japan

* sminamis@jikei.ac.jp

**Data Availability Statement:** All data of DNA microarray analysis are at NCBI GEO (GSE141650). All the other relevant data are within the the manuscript and its Supporting Information files.

## Abstract

Myocardial fibrosis is often associated with cardiac hypertrophy; indeed, fibrosis is one of the most critical factors affecting prognosis. We aimed to identify the molecules involved in promoting fibrosis under hypertrophic stimuli. We previously established a rat model of cardiac hypertrophy by pulmonary artery banding, in which approximately half of the animals developed fibrosis in the right ventricle. Here, we first comprehensively analyzed mRNA expression in the right ventricle with or without fibrosis in pulmonary artery banding model rats by DNA microarray analysis (GSE141650 at NCBI GEO). The expression levels of 19 genes were up-regulated more than 1.5-fold in fibrotic hearts compared with non-fibrotic hearts. Among them, fibrosis growth factor (FGF) 23 showed one of the biggest increases in expression. Real-time PCR analysis also revealed that, among the FGF receptor (FGFR) family, FGFR1 was highly expressed in fibrotic hearts. We then found that FGF23 was expressed predominantly in cardiomyocytes, while FGFR1 was predominantly expressed in fibroblasts in the rat ventricle. Next, we added FGF23 and transforming growth factor (TGF)-β1 (10–50 ng/mL of each) to isolated fibroblasts from normal adult rat ventricles and cultured them for three days. While FGF23 itself did not directly affect the expression levels of any fibrosis-related mRNAs, FGF23 enhanced the effect of TGF-β1 on increasing the expression levels of α-smooth muscle actin (α-SMA) mRNA. This increase in xx-SMA mRNA levels due to the combination of TGF-β1 and FGF23 was attenuated by the inhibition of FGFR1 or the knockdown of FGFR1 in fibroblasts. Thus, FGF23 synergistically promoted the activation of fibroblasts with TGF-β1, transforming fibroblasts into myofibroblasts via FGFR1. Thus, we identified FGF23 as a paracrine factor secreted from cardiomyocytes to promote cardiac fibrosis under conditions in which TGF-β1 is activated. FGF23 could be a possible target to prevent fibrosis following myocardial hypertrophy.

**Funding:** This work was supported by grants from the Ministry of Education, Culture, Sports, Science and Technology of Japan, the Vehicle Racing Commemorative Foundation, the Uehara Memorial Foundation, and The Jikei University Graduate Student Research Grant. The funders had no role in study design, data collection and analysis, decision to publish, or preparation of the manuscript.

**Competing interests:** The authors have declared that no competing interests exist.

## Introduction

Cardiac fibrosis often follows cardiac hypertrophy induced by pressure overload. It stiffens the heart to provide resistance to pressure overload, but also promotes diastolic dysfunction of the heart. Because cardiac fibrosis is rarely reversible, controlling cardiac fibrosis is important to reduce the risk of heart failure. Studies on the molecular mechanisms of cardiac fibrosis have identified transforming growth factor (TGF)-β1 as one of the most important factors involved in fibrosis. However, TGF-β1 is known to be related to both hypertrophy and fibrosis [1, 2]. Distinguishing between a factor for myocardial fibrosis and a hypertrophic factor is challenging because myocardial fibrosis and cardiac hypertrophy typically occur together.

In the previous study, we established pulmonary artery banding (PAB) model rats which exhibited heavier right ventricular weight and wider right atrium dimension compared to normal rats [3, 4]. Interestingly, approximately half of PAB rats developed fibrosis in the right ventricle, although hypertrophy was observed in all animals at four weeks after PAB operation. Therefore, we considered that this model could enable us to distinguish the factors involved in myocardial fibrosis from those involved in hypertrophy.

The purpose of this study is to clarify the mechanisms of cardiac fibrosis as distinct from those of hypertrophy. We comprehensively analyzed the gene expression profiles in PAB rat right ventricles with or without fibrosis by DNA microarray.

## Materials and methods

### Animals

Experiments were performed after obtaining approval from the Animal Experiment Committee of The Jikei University School of Medicine. Sprague-Dawley (SD) rats were obtained from Sankyo Labo Service Corporation (Japan). Rats were allowed free access to a pelleted laboratory animal diet and tap water.

### PAB model

In our previous study [3] we demonstrated that, four weeks after the PAB operation, rats could be divided into two groups: an F+ group in which the fibrotic area occupied more than 6.5% of the whole area of the heart tissues, and an F- group in which the fibrotic area occupied less than 6.5% of this area. We used heart samples obtained from PAB or sham-operated rats in the present study.

### Microarray analysis

Frozen right ventricles in the Sham, F-, and F+ groups were immersed in 1 mL of TRIzol Reagent (Invitrogen, Carlsbad, CA, USA) and crushed using a bead-type homogenizer. After centrifugation at 12,000 g at 4°C for 15 minutes, the supernatant was collected. Microarray analysis was performed as previously described [5, 6]. Briefly, total mRNA was extracted following the instructions attached to the kit. Sense-strand cDNA containing dUTP was synthesized by amplified cRNA. These fragmented cDNAs (25 μg) were then labeled through a terminal deoxy-transferase reaction and hybridized to the Affymetrix GeneChip® Rat Gene 1.0 ST Array (Affymetrix, Santa Clara, CA, USA). Each array was then washed and stained on the GeneChip fluidics station 450 using the appropriate fluidics script; once completed, the array was inserted into the Affymetrix autoloader carousel and scanned using the GeneChip Scanner 3000. The hybridization experiments were performed in triplicate, and the intensities were averaged.

## Isolation of cardiomyocytes and fibroblasts from adult rat hearts and cell culture

Male SD rats at 9–12 weeks old were supplied for cell isolation. Animals were anesthetized with pentobarbital (100 mg/kg, i.p.). Heparin sodium was injected to prevent blood clotting after anesthetization. The chest cavity was opened and the heart rapidly excised. The excised heart was perfused in a reverse fashion via the aorta with a HEPES Tyrode's solution (137 mM NaCl, 5.4 mM KCl, 0.5 mM $MgCl_2$, 0.3 mM $NaH_2PO_4$, 5 mM HEPES, 0.9 g/L glucose, 2 mM $CaCl_2$, pH7.4) using a Langendorff perfusion apparatus. After the remaining blood was washed out and the heartbeat was stable, the heartbeat was stopped with potassium chloride solution. Next, the heart was perfused with heart media solution (S-MEM [Gibco, Waltham, MA, USA] with 2.4 g/L HEPES, 3.76 g/L Taurin, 0.40 g/L DL-carnitine and 0.3 g/L Creatine, pH7.4) for 6 min, then with collagenase solution (heart media solution with 13000 units/head collagenase L, 1% bovine serum albumin [BSA], 20 μM $CaCl_2$) for 20 min at 37˚C. After perfusion was completed, the ventricle was isolated and cut into small pieces. Cells were triturated with a transfer pipette for 6 min at 37˚C and filtered through a Cell Strainer (100 μm, BD Falcon, San Jose, CA, USA), then incubated in a mixture of 5 mL collagenase solution and 5 mL washing solution (heart media solution with 10% BSA, 20 μM $CaCl_2$) at room temperature for 9 min. Fibroblasts were collected from the supernatant and cardiomyocytes were collected from precipitation.

Supernatant containing fibroblasts was centrifuged (1000 rpm, 10 minutes, r.t.) and precipitation was collected. The fibroblasts were suspended in plating medium (DMEM, High Glucose, GlutaMAX™, Pyruvate [Gibco] with 10% fetal bovine serum [FBS] and 1% penicillin/streptomycin solution) and plated. Cells were cultured at 37˚C with 5% $CO_2$. Cardiomyocytes were washed twice with washing buffer, and 100 mM $CaCl_2$ was added five times every four minutes. Separated cardiomyocytes were dispersed in attaching medium (Medium 199 [Invitrogen] with 4% FBS and 1% penicillin/streptomycin solution) and seeded to laminin-coated dishes. Cells were cultured at 37˚C with 5% $CO_2$ for 1–2 hours in connecting medium. After cells were attached, the cardiomyocyte cells were suspended in maintenance medium (199 Medium with 1% BSA and 1% penicillin/streptomycin solution) and cultured at 37˚C with 5% $CO_2$ for about 24 hours.

## FGF23 and TGF-β1 treatments

At approximately 50% confluence, fibroblasts were treated with TGF-β1 (PeproTech, Rocky Hill, NJ, USA) and FGF23 recombinant protein (R&D Systems, Minneapolis, MN, USA) in low serum medium (DMEM with 1% FBS and 1% penicillin/streptomycin solution) for three days. Dose levels of TGF-β1 and FGF23 were 10, 25 and 50 ng/mL.

To inhibit the activity of FGFR1, 10 μM of SU5402 (R&D Systems) was added to some wells prior to treatment with FGF23 and TGF-β1. Cells were then treated with 10 ng/mL of TGF-β1 and/or 25 ng/mL of FGF23 in low-serum medium for three days.

For other fibroblasts, medium was changed to Opti-MEM (Gibco). Mixture of 3% lipofectamin and 1% siRNA (Fgfr1, GENE_ID 79114, Bioneer Corporation, Daejeon, Korea) was added, and cells were incubated at 37˚C with 5% $CO_2$ overnight. Cells were then treated with 10 ng/mL of TGF-β1 and/or 25 ng/mL of FGF23 in low serum medium for three days.

## Quantitative real-time PCR

Quantitative real-time PCR was performed as previously described [3, 7]. Briefly, heart tissues from PAB or sham operated rats were immersed in 1 mL of TRIzol Reagent (Invitrogen) or

**Table 1. List of primer sequence used for RT-PCR.**

| Gene name | Forward primer | Reverse primer |
|---|---|---|
| GAPDH | TGGTGAAGCAGGCATCTGAG | TGCTGTTGAAGTCGCAGGAG |
| FGF23 | GCCAGGAACAGCTATCACCTACAGA | GTTGCCGCGGAGATCCATAC |
| Klotho | GCAAAGCGCTCAACTGGCTAA | GCGAATACGCAAAGTAGCCACA |
| FGFR1 | CAGGGCTACCAGCCAACAA | CACTGTACACCTTGCACATGAACTC |
| FGFR2 | TGTTTCAACTCTGCTGTCCGATG | CATCTTGGGATGAGGACTCTGGTA |
| FGFR3 | CCCAGAACCCTGACCAAGTA | CCCAGAACCCTGACCAAGTA |
| FGFR4 | CGAGGCATGCAGTATCTGG | CCAAAGTCAGCGATCTTCATCAC |
| α-SMA | AGCCAGTCGCCATCAGGAAC | GGGAGCATCATCACCAGCAA |
| Pro collagen I | CAGCGGAGAGTACTGGATCGA | CTGACCTGTCTCCATGTTGCA |
| Pro collagen III | TGCCATTGCTGGAGTTGGA | GAAGACATGATCTCCTCAGTGTTGA |

GAPDH: glyceraldehyde 3-phosphate dehydrogenase; TGF-β1: transforming growth factor-β1; FGF23: fibroblast growth factor 23; FGFR: FGF receptor; α-SMA: α-smooth muscle actin

Sepasol-RNA Super G (Nacalai Tesque, Kyoto, Japan) and crushed using a bead-type homogenizer. Fibroblasts and cardiomyocytes isolated from normal rat hearts were immersed in 1 mL of TRIzol Reagent (Invitrogen) or Sepasol-RNA Super G (Nacalai Tesque). Total RNA was extracted through sequential treatment with chloroform, 2-propanol, and ethanol. cDNA was synthesized using a TAKARA PCR Thermal Cycler Dice ™ (Takara Bio, Shiga, Japan), and real-time PCR was performed using a Thermal Cycler Dice 1 and SYBR 1 Premix Ex Taq™ (Takara Bio). The nucleotide sequences of the primers used are shown in Table 1. The experiments were performed in duplicate or triplicate, and the intensities were averaged. The GAPDH mRNA expression level was quantitated as an internal reference.

### Immunohistological analysis

Fibroblasts were fixed with 4% paraformaldehyde 72 hours after treatment with 10 ng/mL TGF-β1 and 25 ng/mL FGF23. Fixed cells were stained with primary antibody (1% of α-SMA [ab7817, Abcam, Cambridge, UK] and 0.4% of vimentine [ab92547, Abcam] in TPBS/BSA solution) at 4˚C overnight. Samples were then incubated in a goat anti-mouse IgG secondary antibody for 1 hr at room temperature. The slides were rinsed in DPBS and counterstained with 4,6-diamidino-2-phenylindole (DAPI) in mounting medium (Santa Cruz Biotechnology, Santa Cruz, CA, USA). Images were captured on a fluorescence microscope (Keyence, Osaka, Japan)

### Statistical analysis

Data are presented as mean ± standard error (SEM) of independent experiments. Statistical analyses were performed between two groups by Wilcoxon–Mann-Whitney U test and among multiple groups by one-way analysis of variance (ANOVA) followed by Kruskal-Wallis test. A p value of <0.05 was considered significant.

## Results

### FGF23 mRNA expression was upregulated in PAB rat fibrotic heart

We comprehensively analyzed the expression profiles of 29215 rat genes in the right ventricles of control and PAB rats using the Affymetrix GeneChip® Rat Gene 1.0 ST Array (Affymetrix). The expression levels of 19 genes in the F+ group were up-regulated more than 1.5-fold

relative to their levels in the F- group (Table 2). Among these genes, we focused on fibrosis growth factor (FGF) 23 because a previous study had reported that FGF23 plays an important role in cardiac hypertrophy [8]. We found, however, that the expression levels of FGF23 mRNA in the F- group were equivalent to those in the sham-operated group, suggesting that hypertrophy itself is not a key factor in increasing the expression levels of FGF23 mRNA in the rat heart. PAB did not change the expression levels of other members of the FGF superfamily (S1 Fig). The FGF23 up-regulation in the F+ group was also confirmed through real-time PCR (Fig 1A). Although no clear changes in any FGF receptors (FGFRs) were observed through microarray analysis, the expression levels of FGFR1 mRNA were significantly higher in the F + group compared to the F- group in RT-PCR analysis (Fig 1B). In contrast, the expression levels of FGFR2, FGFR3 and FGFR4 were comparable between the F+ group and the F- group (Fig 1B). Levels of α-Klotho, which is known to work as an FGF23 receptor with FGFR1 in the kidney, were comparable among the three groups (Fig 1C).

## FGF23 mRNA is more highly expressed in fibroblasts than in cardiomyocytes

We then sought to determine which cell type could be responsible for the up-regulation of FGF23 mRNA in the cardiomyocytes and fibroblasts that were isolated from adult rat ventricles. Real-time PCR analysis revealed that the expression levels of FGF23 mRNA were significantly higher in cardiomyocytes than in fibroblasts (Fig 2A). In contrast, the expression levels of FGFR1 mRNA were significantly lower in cardiomyocytes than in fibroblasts (Fig 2B).

**Table 2. Fold changes of gene expression levels.**

| Gene symbol | F-/Sham | F+/Sham | F+/F- |
|---|---|---|---|
| FMGC72973 | 0.9 | 1.77 | 1.97 |
| Hbb | 0.86 | 1.63 | 1.89 |
| Olr711 | 0.86 | 1.61 | 1.87 |
| Hbb/Hbb | 0.89 | 1.64 | 1.85 |
| RGD1560242 | 1.15 | 2.11 | 1.83 |
| Nppa | 0.93 | 1.67 | 1.79 |
| Thbs4 | 1.32 | 2.2 | 1.66 |
| Mcpt2 | 0.99 | 1.61 | 1.63 |
| Cpa3 | 0.99 | 1.6 | 1.61 |
| Serpina3n | 0.98 | 1.56 | 1.6 |
| RGD1559459 | 0.97 | 1.55 | 1.6 |
| Eraf | 0.89 | 1.41 | 1.59 |
| Olr1598 | 0.84 | 1.34 | 1.58 |
| Olr850 | 1.16 | 1.84 | 1.58 |
| Ncam1 | 1.15 | 1.81 | 1.58 |
| **Fgf23** | **1.01** | **1.59** | **1.57** |
| Tmem119 | 0.96 | 1.48 | 1.54 |
| Hba-a2 | 0.87 | 1.35 | 1.54 |
| Olr1418 | 0.73 | 1.1 | 1.52 |

Microarray analysis in right ventricular without fibrosis (F-) or with fibrosis (F+). F-/sham: Fold changes of F- from sham-operated group. F+/sham: Fold changes of F+ from sham-operated group. F+/F-: Fold changes of F+ from F- group.

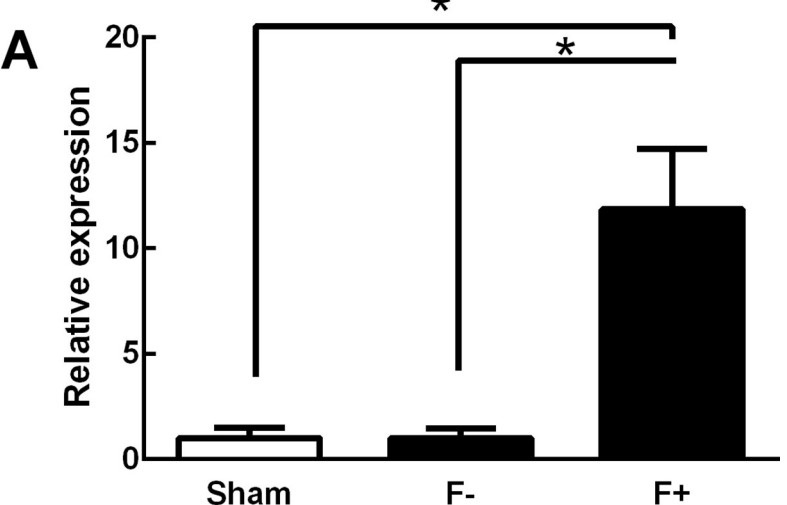

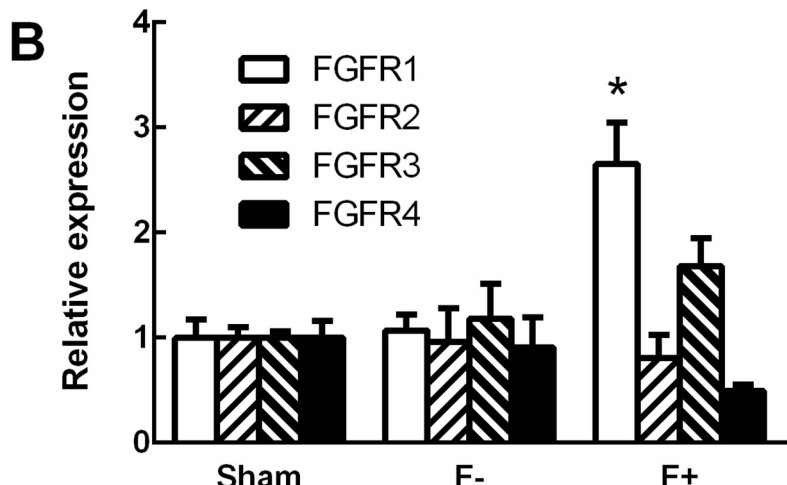

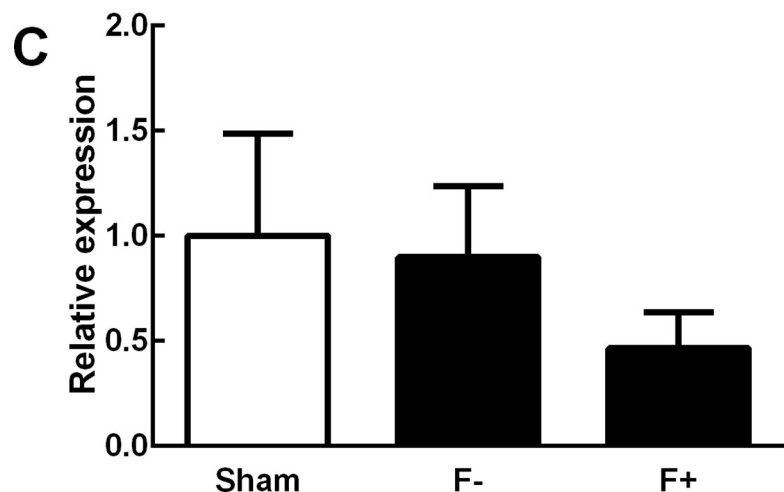

**Fig 1. Relative mRNA expression levels in right ventricles of pulmonary artery banding model rats.** (A) Expression of FGF23. (B) Expression of FGFR1, 2, 3 and 4. (C) Expression of Klotho. Values are means ± SEM; n = 4–6 for each.

### FGF23 promoted myofibroblast transformation in the presence of TGF-β1

Cultured fibroblasts from adult rat ventricles were treated with TGF-β1 and/or FGF23 for three days. We found that TGF-β1 tended to increase the expression levels of α-SMA mRNA in a dose-dependent manner (Fig 3A), whereas FGF23 did not change the expression levels of α-SMA mRNA up to 50 ng/mL (Fig 3B). We then examined the synergic effect of FGF23 on TGF-β1-induced α-SMA up-regulation. In the presence of 25 ng/mL FGF23 in combination with 25 ng/mL of TGF-β1, the expression levels of α-SMA mRNA were increased (Fig 3C). On the other hand, the expression levels of pro collagen I and III, were not changed in the presence of 25 ng/mL FGF23 in combination with 10 ng/mL of TGF-β1 (Fig 3D and 3E). In the presence of FGF23, α-SMA expression reached maximal levels even with only 10 ng/mL TGF-β1; these α-SMA levels were comparable with those seen in the 50 ng/mL TGF-β1 (Fig 4). Immunohistological analysis revealed that FGF23 alone did not affect the morphology or α-SMA expression of fibroblasts (Fig 5A and 5B). Nevertheless, TGF-β1 treatment clearly increased the size of each fibroblast and tended to increase the ratio of α-SMA-positive cells (Fig 5C). Moreover, α-SMA-positive cells were further increased in fibroblasts treated with both TGF-β1 and FGF23 compared with those treated with TGF-β1 alone (Fig 5D).

### FGF23-induced α-SMA up-regulation was suppressed by FGFR1 inhibition

The FGFR1 inhibitor SU5402 decreased the α-SMA mRNA expression level in fibroblasts treated with both TGF-β1 and FGF23, but did not affect fibroblasts in the absence of any treatment or in the presence of TGF-β1 alone (Fig 6A). Down-regulation of FGFR1 by siRNA also tended to attenuate the expression levels of α-SMA mRNA in fibroblasts treated with both TGF-β1 and FGF23 (Fig 6B).

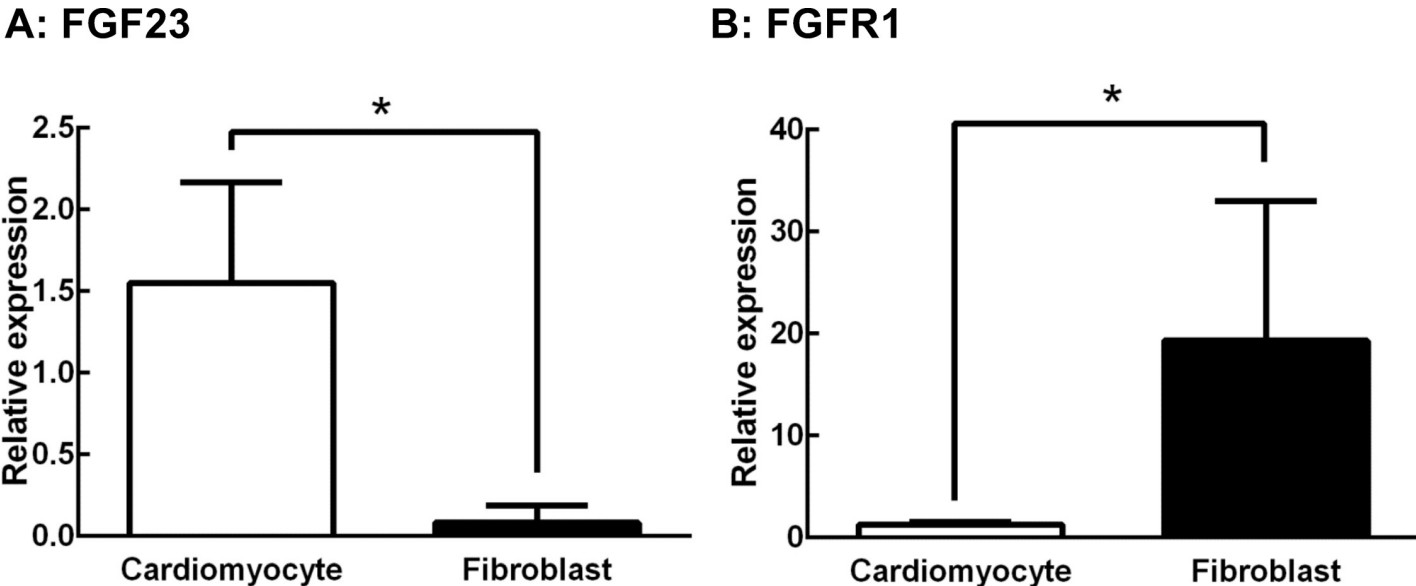

**Fig 2. Relative mRNA expression levels in cardiomyocytes and fibroblasts as measured by RT-PCR.** (A) Expression of FGF23. (B) Expression of FGFR1. Values are means ± SEM; n = 6 for each. (*) $P < 0.05$.

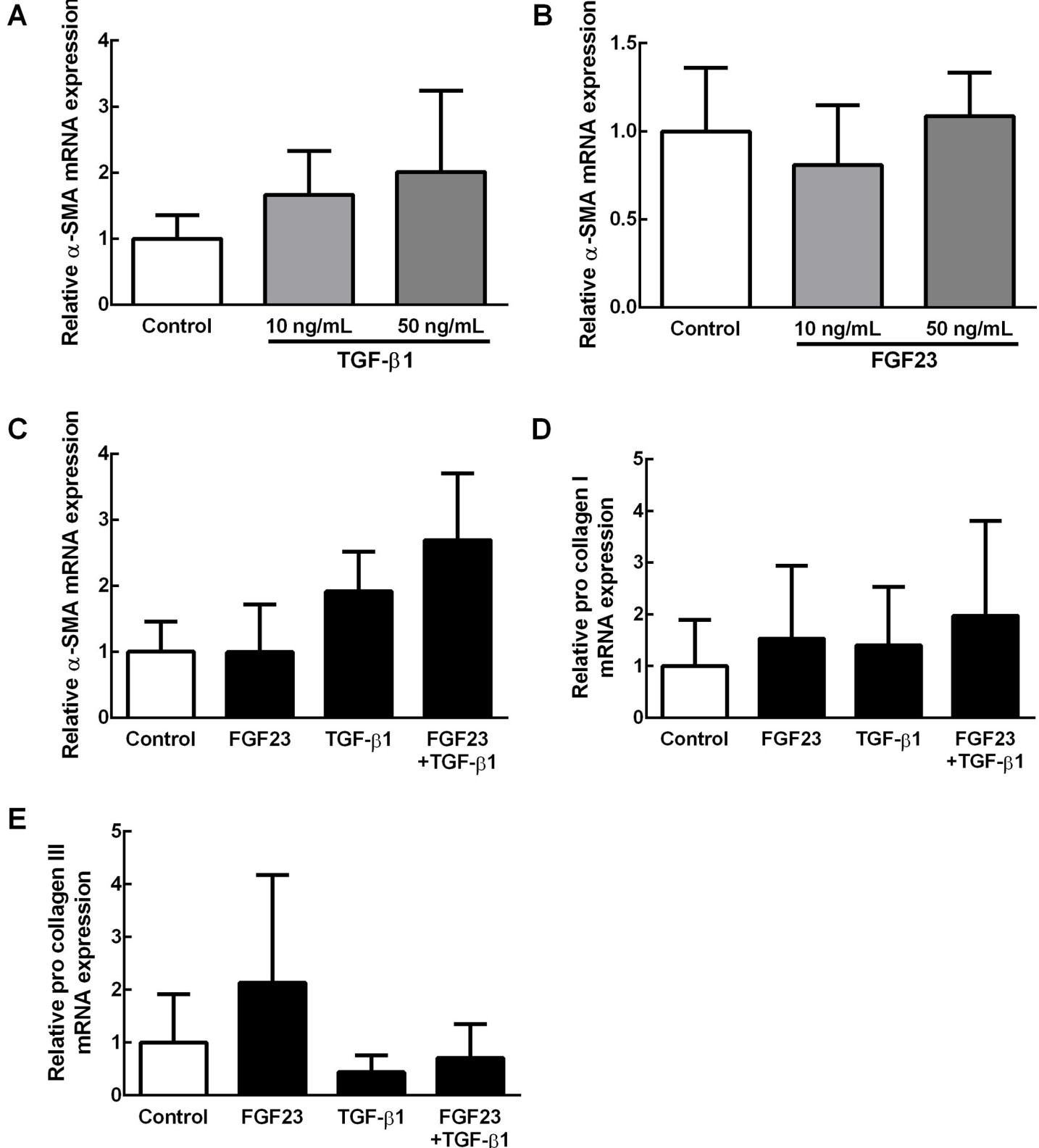

**Fig 3. mRNA expression levels in fibroblasts as measured by RT-PCR.** (A) α-SMA expression levels after treatment with TGF-β1 (n = 6). (B) α-SMA expression levels after treatment with FGF23 (n = 6). (C) α-SMA expression levels after treatment with combination of FGF23 and TGF-β1 (n = 5). (D) pro collagen I expression levels after treatment with a combination of FGF23 and TGF-β1 (n = 5). (E) pro collagen III expression levels after treatment with a combination of FGF23 and TGF-β1 (n = 5), Values are means ± SEM.

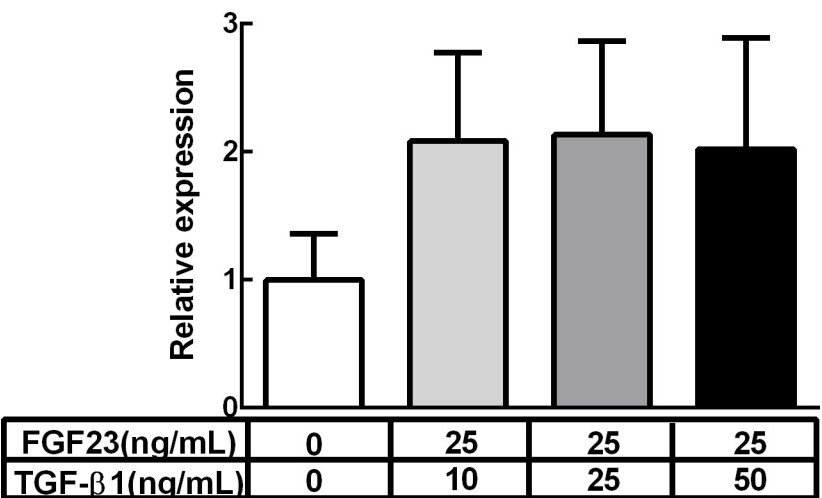

**Fig 4. α-SMA mRNA expression levels in fibroblasts treated with FGF23 and various doses of TGF-β1.** Relative α-SMA mRNA expression levels 3 days after treatment with a combination of 25 ng/mL FGF23 and various doses of TGF-β1 as measured by RT-PCR. Values are means ± SEM. n = 5 for each.

## Discussion

The most significant finding of the present study is that FGF23 can be induced from cardio-myocytes in the rat heart under a profibrotic condition. Our study indicates that hypertrophic stimuli may not be sufficient to induce FGF23 in the heart, although a number of previous studies have demonstrated that FGF23 is associated with myocardial hypertrophy [8–10]. In this regard, Slavic et al. demonstrated that genetic deletion of FGF23 did not affect the patho-physiology of pressure overload-induced cardiac hypertrophy [11]. Hao et al. were the first, to our knowledge, to demonstrate that FGF23 instead promotes myocardial fibrosis in mice through the activation of β-catenin [12]. They showed that FGF23 promotes proliferation, col-lagen I and III synthesis, and β-catenin activation in an ischemic condition. Importantly, they suggested that endogenous cardiac FGF23 promotes myocardial fibrosis after myocardial infarction or ischemia reperfusion, but not under normal conditions, through induction of paracrine signaling pathways. Furthermore, our results are consistent with those of a recent study by Leifheit-Nestler et al. [13] demonstrating that FGF23 induced from cardiac myocytes promoted cardiac fibrosis via the profibrotic crosstalk between cardiac myocytes and fibro-blasts. Although Leifheit-Nestler et al. indicated that FGF23 itself works as a profibrotic factor [13], our results suggested that FGF23 itself did not have profibrotic action but rather synergis-tically activated fibroblasts in the presence of TGF-β1, which is known to induce myofibroblast trans-differentiation via the Smad-3 and Wnt signaling pathways [14]. It should be noted that Leifheit-Nestler et al. found that the profibrotic effects of FGF23 were weaker than those of TGF-β1 and that they did not examine the possibility of a synergic effect between FGF23 and TGF-β1 [13]. In keeping with Hao et al.'s study [12], we assume that FGF23 promotes fibrosis under profibrotic conditions such as ischemia or hypoxia. Importantly, Leifheit-Nestler et al. demonstrated that FGF23 was elevated in the hearts of human patients with end-stage chronic kidney disease through a comparative analysis of fibrosis-related gene expression profiling in human myocardial tissues [13]. Although trans-differentiation to myofibroblast was suggested, the expression levels of pro collagen I and III mRNAs were not clearly changed in our study. The expression levels of collagen have been reported to be increased early in response to

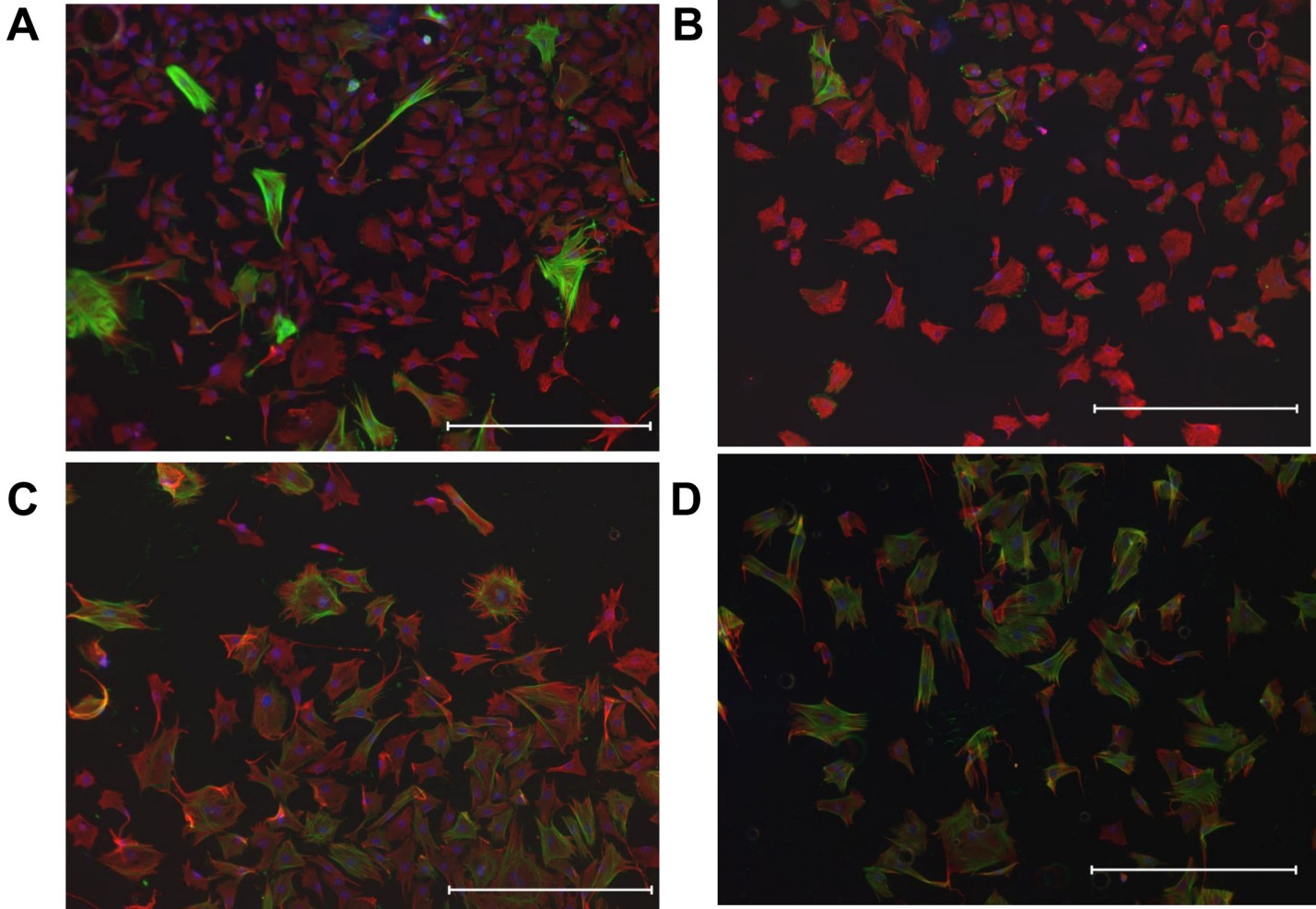

**Fig 5. Immunohistological staining of fibroblasts.** (A) Control. (B) After treatment with 25 ng/mL FGF23. (C) After treatment with 10 ng/mL TGF-β1. (D) After treatment with both 25 ng/mL FGF23 and 10 ng/mL TGF-β1. Red: α-SMA; Green: vimentine; Blue: nuclear staining with 4, 6-diamidino-2-phenylindole (DAPI). Scale bar indicates 500 μm.

stimuli and then decreased [15, 16]. It is possible that the expression levels of collagen have already peaked out as we measured them at 72 hours after treatments.

FGF23 is mainly secreted from osteoblasts and has an endocrine effect on the kidney through activation of FGFR1/klotho co-receptor complexes to regulate phosphate and mineral homeostasis [17]. In addition to its main action, the role of FGF23 in cardiac dysfunction is also attracting considerable attention because an increase in plasma FGF23 levels is associated with the risk of heart failure and circulating FGF23 is considered a possible biomarker for heart failure [18–20]. FGF23 has also been proposed to be secreted from hearts [21, 22]. Moreover, we found that FGF23 is highly expressed in cardiomyocytes compared with fibroblasts. This suggests that FGF23 has a paracrine effect on myofibroblasts in which FGFR1 is highly expressed. In support of that hypothesis, we demonstrated that FGFR1 inhibition or down-regulation affects FGF23 function to promote cardiac fibrosis. Furthermore, we found that the expression levels of FGFR1 mRNA were significantly higher in the fibrotic hearts as measured by RT-PCR analysis, but not as measured by DNA microarray analysis. This discrepant result between RT-PCR and DNA microarray analyses could be due to the lower detection sensitivity

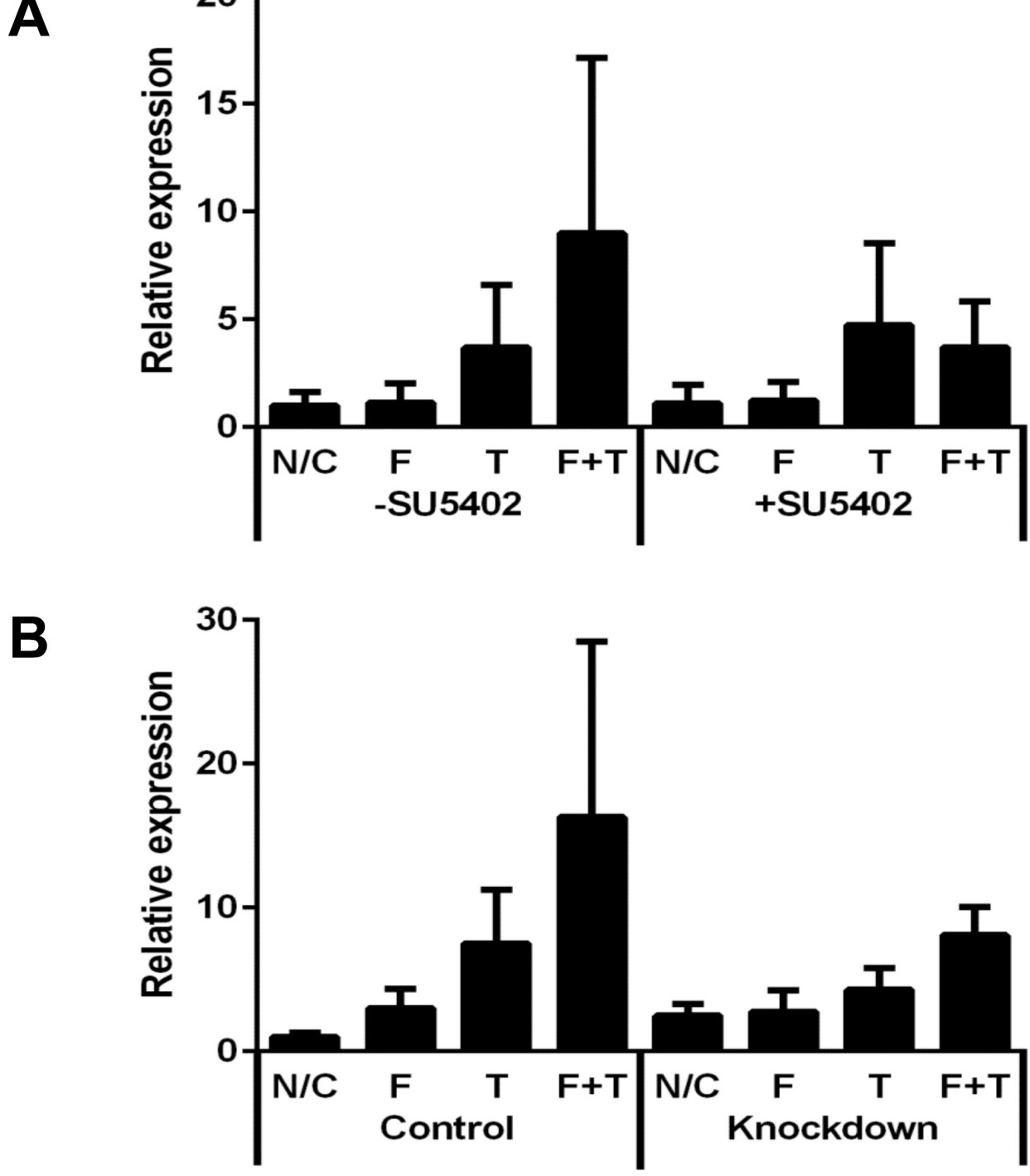

**Fig 6. Relative α-SMA mRNA expression levels in fibroblasts after inhibition or down-regulation of FGFR1.** Relative α-SMA mRNA expression levels in fibroblasts 3 days after treatment with FGF23 and/or TGF-β1. (A) After treatment with FGFR1 inhibitor SU5402 (n = 5). (B) After down-regulation of FGFR1 (n = 4). N/C: Negative control; F: 25 ng/mL FGF23; T: 10 ng/mL TGF-β1. Values are means ± SEM.

of DNA microarray analysis. In keeping with this finding, the paracrine effect of FGF23 has already been demonstrated in other organs such as the kidney [23, 24], vessels [25], and endothelial cells [26].

It would be intriguing to examine how FGF23 could be induced in a profibrotic condition. The present study suggests that hypertrophic stimuli are not sufficient to up-regulate FGF23 mRNA expression levels. Leifheit-Nestler et al. recently demonstrated that the activation of renin-angiotensin II-aldosterone signaling plays an important role in FGF23 induction in the heart and that both signaling pathways synergistically contribute to cardiac remodeling [13]. Although we did not observe any changes in the mRNA expression levels of genes related to renin-angiotensin II-aldosterone signaling in our DNA microarray analysis, further study is required to clarify whether renin-angiotensin II-aldosterone signaling is the sole factor inducing FGF23 in cardiomyocytes. In this regard, our DNA microarray analysis revealed the up-regulation of other genes whose roles in fibrosis have not been examined. Therefore, investigating the interactions between FGF23 and these genes would be an interesting future study.

In conclusion, using an animal model that clearly distinguishes between non-fibrotic and fibrotic hypertrophy in the right ventricle, we identified FGF23 as a paracrine factor secreted from cardiomyocytes to promote cardiac fibrosis under a condition in which TGF-β1 is activated. FGF23 could be a possible target of treatments intended to prevent fibrosis following myocardial hypertrophy.

## Supporting information

**S1 Fig. Fold changes of gene expression levels in FGF families.** Microarray analysis in right ventricular without fibrosis (F-) or with fibrosis (F+).
(TIF)

## Acknowledgments

This work was supported by grants from the Ministry of Education, Culture, Sports, Science and Technology of Japan, the Vehicle Racing Commemorative Foundation, the Uehara Memorial Foundation, and The Jikei University Graduate Student Research Grant.

## Author Contributions

**Conceptualization:** Kazuhiro Kuga, Yoichiro Kusakari, Ken Uesugi, Susumu Minamisawa.

**Data curation:** Kazuhiro Kuga.

**Formal analysis:** Kazuhiro Kuga, Yoichiro Kusakari, Ken Uesugi.

**Funding acquisition:** Yoichiro Kusakari, Susumu Minamisawa.

**Investigation:** Kazuhiro Kuga, Ken Uesugi, Takashi Urashima.

**Methodology:** Yoichiro Kusakari, Ken Uesugi, Takashi Urashima, Toru Akaike.

**Project administration:** Kazuhiro Kuga, Yoichiro Kusakari, Susumu Minamisawa.

**Resources:** Kentaro Semba, Takashi Urashima, Toru Akaike, Susumu Minamisawa.

**Supervision:** Kentaro Semba, Susumu Minamisawa.

**Validation:** Kazuhiro Kuga.

**Visualization:** Kazuhiro Kuga.

**Writing – original draft:** Kazuhiro Kuga.

**Writing – review & editing:** Kazuhiro Kuga, Yoichiro Kusakari, Ken Uesugi, Kentaro Semba, Takashi Urashima, Toru Akaike, Susumu Minamisawa.

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
