## [Decision Letter · Decision Letter 0]

20 Jan 2020

PONE-D-19-34439

Fibrosis growth factor 23 is a promoting factor for cardiac fibrosis in the presence of transforming growth factor-β1

PLOS ONE

Dear Prof Minamisawa,

Thank you for submitting your manuscript to PLOS ONE. After careful consideration, we feel that it has merit but does not fully meet PLOS ONE’s publication criteria as it currently stands. Therefore, we invite you to submit a revised version of the manuscript that addresses the points raised during the review process.

We would appreciate receiving your revised manuscript by Mar 05 2020 11:59PM. To enhance the reproducibility of your results, we recommend that if applicable you deposit your laboratory protocols in protocols.io, where a protocol can be assigned its own identifier (DOI) such that it can be cited independently in the future. For instructions see: http://journals.plos.org/plosone/s/submission-guidelines#loc-laboratory-protocols

We look forward to receiving your revised manuscript.

Kind regards,

Michael Bader

Academic Editor

PLOS ONE

Journal Requirements:

Please ensure that your manuscript meets PLOS ONE's style requirements, including those for file naming. The PLOS ONE style templates can be found at http://www.plosone.org/attachments/PLOSOne_formatting_sample_main_body.pdf and http://www.plosone.org/attachments/PLOSOne_formatting_sample_title_authors_affiliations.pdf

Reviewers' comments:

Reviewer's Responses to Questions

**Comments to the Author**

1. Is the manuscript technically sound, and do the data support the conclusions?

Reviewer #1: Yes

Reviewer #2: Yes

2. Has the statistical analysis been performed appropriately and rigorously? 

Reviewer #1: Yes

Reviewer #2: Yes

3. Have the authors made all data underlying the findings in their manuscript fully available?

Reviewer #1: Yes

Reviewer #2: Yes

4. Is the manuscript presented in an intelligible fashion and written in standard English?

Reviewer #1: Yes

Reviewer #2: Yes

5. Review Comments to the Author

Reviewer #1: It is well-known that FGF23 induces LVH, but the molecular mechanism remains unclear. In this manuscript, the authors have identified that FGF23 as a paracrine factor secreted from cardiomyocytes to promote cardiac fibrosis under conditions in which TGF-β1 is activated. It is an interesting article in the field. There are a few of concerns about the manuscript.

Major Concern:

1. What is the level of serum FGF23 (including Sham, F-, and F+ groups) in the rat model of cardiac hypertrophy by pulmonary artery banding compared with control rat?

2. What is the level of serum Klotho (including Sham, F-, and F+ groups) in the rat model of cardiac hypertrophy by pulmonary artery banding compared with control rat?

3. Are there any differences of FGF23, FGFR1, 2, 3 and 4, and klotho expressions in fibroblasts verse in cardiomyocytes from Sham, F-, and F+ groups?

4. A panel of fibrosis- or EMT-related gene expressions should be examined in Figure 3, 4 and 6 by real-time RT-PCR analysis, such as Collagen I, TGF-β, α-SMA, Vimentin, and Snail1.

Minor Concern:

1. In Table 2, the authors are suggested to clarify the experimental groups (F-/Sham? F+/Sham? and F+/F-?).

2. English editing is needed for publication.

Reviewer #2: In the manuscript "Fibrosis growth factor 23 is a promoting factor for cardiac fibrosis in the presence of transforming growth factor-β" by Kazuhiro Kuga et al，the authors investigate that FGF23 and FGFR1 were both highly expressed in the fibrotic hearts of rat. FGF23 synergistically promoted the activation of fibroblasts with TGF-β transforming fibroblasts into myofibroblasts via FGFR1.

The findings are relevant to the field. Although the draft is technically sound with appropriate methods of analyses, several questions raised by reviewers and additional issues have not been fully addressed:

1. As far as this reviewer knows, pulmonary artery banding, which was previously established by the authors, was not really a typical mouse model of cardiac remodeling. Did the authors assessed the FGF23 expression, myocardial hypertrophy and cardiac fibrosis in left ventricle?

2. How did the authors come to the total number of samples? Were power analysis done before experiments? The reason for the variable sample sizes should be explained (Such as Fig. 1, n = 4-6 for each group). Were there technical failures that would otherwise affect interpretation or generalizations of the data?

3. The expression levels of FGFR1 mRNA were significantly higher in the fibrotic hearts. However, no change of FGFR1 was observed through the DNA microarray analysis. Please elaborate the possible reasons in the discussion.

4. Since the right ventricle of rat were used for the DNA microarray analysis, why the authors still choose to use the whole ventricle but not the the right ventricle of adult rat for cell isolation?

5. In Fig. 4, α-SMA expression reached maximal levels with 10 ng/mL TGF-β. Is there a dose-dependent effect of TGF-β on α-SMA expression in fibroblasts by using lower dose levels of TGF-β?

6. For Fig. 1C, Fig. 3, Fig. 4 and Fig. 6, are any of these statistically significant? Authors should provide accurate P values for all statistical analyses.

7. Line 176: “…reported that FGF23 plays an important role in cardiac hypertrophy”, where are the references?

6. PLOS authors have the option to publish the peer review history of their article (what does this mean?). If published, this will include your full peer review and any attached files.

Reviewer #1: No

Reviewer #2: No

---

## [Author Response · Author response to Decision Letter 0]

25 Feb 2020

Dr. Joerg Heber

Editor-in-Chief, PLoS One

Dear Dr. Heber,

We are grateful for the opportunity to revise our manuscript (PONE-D-19-34439), titled “Fibrosis growth factor 23 is a promoting factor for cardiac fibrosis in the presence of transforming growth factor-beta” and for the reviewers’ helpful comments. We were delighted to learn that you and the reviewers consider our work to be important. We have responded to each of the reviewers’ comments in detail (please see our “Responses to the reviewers’ comments”), and have modified the manuscript accordingly. 

There is one thing for which we must apologize to the editor and the reviewers regarding the unclear explanation of our statistical analysis methods in the previous version. Given the opportunity to reexamine our statistical analysis, we reconsidered the methods we had chosen. Since the sample size was small and the data were not normally distributed, we now consider that a non-parametric test would be more appropriate for our research. Therefore, we reevaluated our data using the Wilcoxon–Mann-Whitney U test or one-way analysis of variance (ANOVA) followed by the Kruskal-Wallis test. Despite this reevaluation, our results were not changed. 

It is our hope that the present version satisfies all of the editor’s and reviewers’ requests. The authors would like to sincerely thank you and the reviewers for your time and commitment. We believe that the revised version of our manuscript is much improved, and it is our hope that it will be deemed suitable for publication in your esteemed journal. 

Best regards,

Susumu Minamisawa

Point-by-point responses to the reviewers’ comments

Response to Reviewer 1: 

We thank Reviewer 1 for his/her thorough review of the manuscript and positive comments on our findings. We have responded to each of Reviewer 1’s criticisms and have modified the manuscript accordingly. It is our hope that the revised version is now deemed acceptable. 

Major concern

1. What is the level of serum FGF23 (including Sham, F-, and F+ groups) in the rat model of cardiac hypertrophy by pulmonary artery banding compared with control rat?

Unfortunately, we did not collect blood from PAB model rats because we only noted the relation between FGF23 and fibrosis after PAB model rats had been sacrificed. Moreover, we think that the paracrine mechanism of FGF23 is important in the present study. Therefore, we think that the measurement of serum FGF23 levels is not critical for our study. We appreciate Reviewer 1's suggestion and would like to examine the serum FGF23 levels in PAB model rats in our future research. 

2. What is the level of serum Klotho (including Sham, F-, and F+ groups) in the rat model of cardiac hypertrophy by pulmonary artery banding compared with control rat?

Serum Klotho was not measured in PAB model rats for the same reasons mentioned in comment 1. The physiological role of serum Klotho in the heart is unclear, and we do not suspect a mechanism mediated by Klotho. Therefore, we think that Klotho measurement is not critical for our study.

3. Are there any differences of FGF23, FGFR1, 2, 3 and 4, and klotho expressions in fibroblasts verse in cardiomyocytes from Sham, F-, and F+ groups?

The expression levels of FGF23, FGFRs and Klotho in PAB rats were not measured separately in cardiomyocytes or fibroblasts, and unfortunately there are no remaining samples of PAB rat hearts. It is possible that the expression levels of FGF23 and FGFR1 are different in PAB rats, and we would like to check this point in our future research. Thank you for this useful comment.

4. A panel of fibrosis- or EMT-related gene expressions should be examined in Figure 3, 4 and 6 by real-time RT-PCR analysis, such as Collagen I, TGF-β, α-SMA, Vimentin, and Snail1.

Thank you for your comment. We have added the results of RT-PCR analysis of pro collagen I and III to Fig 3 and added the relevant primer information to Table 1. The expression levels of pro collagen I and III mRNAs were not apparently changed by TGF-�1 and/or FGF23. To clarify the mechanism by which FGF23 contributes to fibrosis, further research is definitely needed. 

Minor concern

1. In Table 2, the authors are suggested to clarify the experimental groups (F-/Sham? F+/Sham? and F+/F-?).

Thank you for this comment. We have added an explanation of this in the footnote to Table 2. 

2. English editing is needed for publication.

We have had a professional scientific editor who is also a native English speaker carefully review the manuscript a second time. 

Responses to Reviewer 2: 

We would like to thank Reviewer 2 for his/her thorough review of the manuscript and positive comments on our findings. We have responded to each of Reviewer 2’s comments and have modified the manuscript accordingly. It is our sincere hope that the revised version is deemed acceptable. 

General comments

1. As far as this reviewer knows, pulmonary artery banding, which was previously established by the authors, was not really a typical mouse model of cardiac remodeling. Did the authors assessed the FGF23 expression, myocardial hypertrophy and cardiac fibrosis in left ventricle?.

We did not measure FGF23 levels in the left ventricle in PAB rats because there were no cases of hypertrophy or fibrosis in the left ventricles of PAB rats. We agree with Reviewer 2 that this should be the next step in investigating FGF23 levels in a typical animal model of left ventricular remodeling, and we hope to pursue this in our future research.

2. How did the authors come to the total number of samples? Were power analysis done before experiments? The reason for the variable sample sizes should be explained (Such as Fig. 1, n = 4-6 for each group). Were there technical failures that would otherwise affect interpretation or generalizations of the data?

Thank you for the important comment. The number of samples was four in the F- and sham groups examined for FGFR3 in Fig. 1B as well as in the F+ group examined for Klotho in Fig. 1C, while it was six for all other experimental settings in Fig. 1. The reason why not all sample sizes were N=6 was due to technical error. We did not conduct a power analysis before the experiments. 

3. The expression levels of FGFR1 mRNA were significantly higher in the fibrotic hearts. However, no change of FGFR1 was observed through the DNA microarray analysis. Please elaborate the possible reasons in the discussion. 

The magnitude of the changes in this study were relatively small for DNA microarray analysis. The difference in detection sensitivity is one possible reason why no increase in FGFR1 was detected in microarray analysis although an increase was detected in RT-PCR. We now mention this possibility in the Discussion section of the revised manuscript.

4. Since the right ventricle of rat were used for the DNA microarray analysis, why the authors still choose to use the whole ventricle but not the the right ventricle of adult rat for cell isolation?

For the in-vitro experiment using normal adult rats in our laboratory, we expected that the responses to TGF-�1 would not be different between the right and left ventricles. Therefore, we used whole ventricles to obtain the largest possible sample size.

5. In Fig. 4, α-SMA expression reached maximal levels with 10 ng/mL TGF-β. Is there a dose-dependent effect of TGF-β on α-SMA expression in fibroblasts by using lower dose levels of TGF-β?

We checked the α-SMA expression in the presence of 1 ng/mL TGF-�� with 25 ng/mL FGF23 in our preliminary study; α-SMA was still increased to almost the same level observed in the presence of 10 ng/mL TGF-� with 25 ng/mL FGF23, suggesting that a much smaller amount of TGF-�1 would be effective in the presence of FGF23. In future research, we would like to confirm the dose dependency using a wider range of dosages.

6. For Fig. 1C, Fig. 3, Fig. 4 and Fig. 6, are any of these statistically significant? Authors should provide accurate P values for all statistical analyses.

In Figs 1C, 3, 4 and 6, there was no statistical significance (P>0.05). We added information on the P value to the title of Fig 1 as it had previously been omitted.

7. Line 176: “…reported that FGF23 plays an important role in cardiac hypertrophy”, where are the references?

We added the relevant reference (Faul C et. al., J Clin Invest; 2011) on Line 176.

---

## [Decision Letter · Decision Letter 1]

26 Mar 2020

PONE-D-19-34439R1

Fibrosis growth factor 23 is a promoting factor for cardiac fibrosis in the presence of transforming growth factor-β1

PLOS ONE

Dear Prof Minamisawa,

Thank you for submitting your manuscript to PLOS ONE. After careful consideration, we feel that it has merit but does not fully meet PLOS ONE’s publication criteria as it currently stands. Therefore, we invite you to submit a revised version of the manuscript discussing the point still raised by reviewer 1.

We would appreciate receiving your revised manuscript by May 10 2020 11:59PM. To enhance the reproducibility of your results, we recommend that if applicable you deposit your laboratory protocols in protocols.io, where a protocol can be assigned its own identifier (DOI) such that it can be cited independently in the future. For instructions see: http://journals.plos.org/plosone/s/submission-guidelines#loc-laboratory-protocols

We look forward to receiving your revised manuscript.

Kind regards,

Michael Bader

Academic Editor

PLOS ONE

Reviewers' comments:

Reviewer's Responses to Questions

**Comments to the Author**

1. If the authors have adequately addressed your comments raised in a previous round of review and you feel that this manuscript is now acceptable for publication, you may indicate that here to bypass the “Comments to the Author” section, enter your conflict of interest statement in the “Confidential to Editor” section, and submit your "Accept" recommendation.

Reviewer #1: (No Response)

2. Is the manuscript technically sound, and do the data support the conclusions?

Reviewer #1: Partly

3. Has the statistical analysis been performed appropriately and rigorously? 

Reviewer #1: Yes

4. Have the authors made all data underlying the findings in their manuscript fully available?

Reviewer #1: (No Response)

5. Is the manuscript presented in an intelligible fashion and written in standard English?

Reviewer #1: Yes

6. Review Comments to the Author

Reviewer #1: Type I Collagen should be upregulated with othe EMT markers. Why was the Type I collagen not upregulated with alpha-SMA in the experiment? It is not consistent with your hypothesis.

7. PLOS authors have the option to publish the peer review history of their article (what does this mean?). If published, this will include your full peer review and any attached files.

Reviewer #1: No

---

## [Author Response · Author response to Decision Letter 1]

2 Apr 2020

Point-by-point responses to the reviewers’ comments

Response to Reviewer 1: 

We thank Reviewer 1 for his/her thorough review of the manuscript. We have responded to Reviewer 1’s comment and have modified the manuscript accordingly. It is our hope that the revised version is now deemed acceptable. 

1. Type I Collagen should be upregulated with the EMT markers. Why was the Type I collagen not upregulated with alpha-SMA in the experiment? It is not consistent with your hypothesis. 

Thank you for pointing out the important issue. We also think that collagen should be increased as alpha-SMA is up-regulated. Collagen has been reported to be increased early in response to stimuli and then be decreased (*see the references below). We measured the expression levels of mRNAs at 72 hours after stimuli and this might be too late to catch the changes in the type I collagen expression. We have mentioned this possibility in the Discussion section of the revised manuscript and would like to confirm this in our future research.

* Smith RL, Lin J, Trindade MC, Shida J, Kajiyama G, Vu T, et al. Time-dependent effects of intermittent hydrostatic pressure on articular chondrocyte type II collagen and aggrecan mRNA expression. .J Rehabil Res Dev. 2000;37(2):153-61.

* Xiaodong Pan, Zhongpu Chen, Rong Huang, Yuyu Yao, and Genshan Ma. Transforming Growth Factor β1 Induces the Expression of Collagen Type I by DNA Methylation in Cardiac Fibroblasts. PLoS One. 2013; 8(4): e60335. doi: 10.1371/journal.pone.0060335

---

## [Editor Report · Decision Letter 2]

3 Apr 2020

Fibrosis growth factor 23 is a promoting factor for cardiac fibrosis in the presence of transforming growth factor-β1

PONE-D-19-34439R2

Dear Dr. Minamisawa,

We are pleased to inform you that your manuscript has been judged scientifically suitable for publication and will be formally accepted for publication once it complies with all outstanding technical requirements.

With kind regards,

Michael Bader

Academic Editor

PLOS ONE
---

## [Editor Report · Acceptance letter]

9 Apr 2020

PONE-D-19-34439R2 

Fibrosis growth factor 23 is a promoting factor for cardiac fibrosis in the presence of transforming growth factor-β1 

Dear Dr. Minamisawa:

I am pleased to inform you that your manuscript has been deemed suitable for publication in PLOS ONE. Congratulations! Your manuscript is now with our production department. 

With kind regards,

on behalf of

Prof. Michael Bader 

Academic Editor

PLOS ONE